# Interlaboratory Study to Evaluate a Testing Protocol for the Safety of Food Packaging Coatings

**DOI:** 10.3390/toxics11020156

**Published:** 2023-02-07

**Authors:** Maricel Marin-Kuan, Vincent Pagnotti, Amaury Patin, Julie Moulin, Helia Latado, Jesús Varela, Yves-Alexis Hammel, Thomas Gude, Heidi Moor, Nick Billinton, Matthew Tate, Peter Alexander Behnisch, Harrie Besselink, Heather Burleigh-Flayer, Sander Koster, David T. Szabo

**Affiliations:** 1Société des Produits Nestlé SA-Nestlé Research, 1000 Lausanne, Switzerland; 2PPG, Pittsburgh, PA 15275, USA; 3SQTS Swiss Quality Testing Services, 8953 Dietikon, Switzerland; 4Gentronix Limited, Alderley Park, Cheshire SK10 4TG, UK; 5BioDetection Systems B.V. (BDS), 1098 XH Amsterdam, The Netherlands

**Keywords:** packaging safety, food contact material, guidance packaging safety, NIAS

## Abstract

According to European regulations, migration from food packaging must be safe. However, currently, there is no consensus on how to evaluate its safety, especially for non-intentionally added substances (NIAS). The intensive and laborious approach, involving identification and then quantification of all migrating substances followed by a toxicological evaluation, is not practical or feasible. In alignment with the International Life Sciences Institute (ILSI) and the European Union (EU) guidelines on packaging materials, efforts are focused on combining data from analytics, bioassays and in silico toxicology approaches for the risk assessment of packaging materials. Advancement of non-targeted screening approaches using both analytical methods and in vitro bioassays is key. A protocol was developed for the chemical and biological screening of migrants from coated metal packaging materials. This protocol includes guidance on sample preparation, migrant simulation, chemical analysis using liquid chromatography (LC-MS) and validated bioassays covering endocrine activity, genotoxicity and metabolism-related targets. An inter-laboratory study was set-up to evaluate the consistency in biological activity and analytical results generated between three independent laboratories applying the developed protocol and guidance. Coated packaging metal panels were used in this case study. In general, the inter-laboratory chemical analysis and bioassay results displayed acceptable consistency between laboratories, but technical differences led to different data interpretations (e.g., cytotoxicity, cell passages, chemical analysis). The study observations with the greatest impact on the quality of the data and ultimately resulting in discrepancies in the results are given and suggestions for improvement of the protocol are made (e.g., sample preparation, chemical analysis approaches). Finally, there was agreement on the need for an aligned protocol to be utilized by qualified laboratories for chemical and biological analyses, following best practices and guidance for packaging safety assessment of intentionally added substances (IAS) and NIAS to avoid inconsistency in data and the final interpretation.

## 1. Introduction

Food contact materials are composed of several substances with the potential to migrate into food and impact human health. Migrating substances may include intentionally added substances (IAS) and non-intentionally added substances (NIAS). IAS are known substances used in the manufacture of the coating while NIAS are often unknown substances found in the coating [1]. 

The hazards of IAS (or known ingredients) can be identified by utilizing existing toxicology information or by conducting in vitro and in vivo toxicology tests. Safe levels of IAS can then be determined and in combination with estimated exposure levels, the risk can be assessed and the same should be applied for NIAS (EFSA 2016). 

NIAS may be either (1) anticipated or predicted substances based on chemical reactions known to occur in the manufacture of food contact materials and/or (2) completely unknown substances that may include impurities, degradation products, or other reaction products [2,3]. The NIAS that cannot be anticipated or predicted are often completely unknown analytically (identity and quantity) and likely lack a toxicological profile. Grob et al. (2006) [4] stated that unknown NIAS can represent more than half of the substances found in the migrant mixture. 

The European Union Framework Regulation states that the safety of food contact materials (FCMs) must be evaluated due to the possible migration of substances into food and that the transfer of these substances to food shall not endanger human health. In 2011, the EU Commission Regulation [5] first mentions the term NIAS regarding plastic materials and articles in contact with food. To ensure safety, the regulation states that NIAS must be assessed by the manufacturer through the scientific principles of risk assessment. To address safety, an ILSI expert group published the guidance documents [6]. 

The European BIOSAFEPAPER project [7,8,9] was an early effort that assessed the safety of NIAS in food contact materials (specifically paper and paperboard). This project combined both chemical and biological analyses and ultimately provided a scientific basis for the use of bioassays in the assessment of other food contact materials. However, the analytical work proposed in this project was very extensive and the determination of safety for the migrant mixture was unclear. 

In 2014, Koster et al. [10] developed a complex mixture safety assessment strategy (CoMSAS) for carton FCMs that incorporated chemical analyses, bioassays and the threshold of toxicological concern (TTC) concept [11,12,13] to assess the unknown NIAS in the migration mixture. The TTC approach utilized an exposure threshold of 90 µg/person/day, once ruling out genotoxicity, likely requiring less identification through analytical analyses while still evaluating safety at low levels [14]. 

In 2015, the European International Life Sciences Institute (ILSI) published guidance on best practices for assessing the safety of NIAS that could be applied to all food contact materials and articles including some elements of the strategy proposed by Koster et al. [6]. Methods and tools were identified for chemical analyses; however, the guidance indicates that it was not feasible to identify, quantify, or assess every NIAS in FCMs and instead suggested approaches that provided a practical solution. Bioassays assessing the toxicological endpoints of genotoxicity, cytotoxicity and endocrine activity were proposed to allow for detecting hazards originating from unidentified NIASs. 

The EFSA scientific committee released a statement addressing the role of genotoxicity assessment for complex mixtures such as FCMs [15]. The committee stated in 2019 that complex mixtures should be chemically characterized as much as possible using state-of-the-art analytical tools. Furthermore, if a mixture contained a fraction of substances that were not chemically identified, the mixture should be concentrated as much as possible and tested in in vitro assays. If the in vitro genotoxicity testing of the whole mixture produces a negative result, then the mixture can be of no concern in regard to genotoxicity and no further testing is needed. 

In 2019, subsequently, the role of specific genotoxicity bioassays was reviewed by an ILSI expert group of the Packaging Materials Task Force [16]. For the detection of direct DNA-reactive mutagens, the Ames test [17] was the recommended bioassay due to its high efficiency and accuracy. However, the Ames test was developed for single-substance testing and has a limited sensitivity for detecting minor constituents in a mixture and consequently without the required TTC detection limit to identify mutagens potentially present in food contact materials. Overall, the authors concluded that the utilization of the TTC concepts along with analytical analyses and bioassays brings valuable benefits to the safety assessment of FCMs and is particularly important as a tool for manufacturers developing new food contact materials. However, efforts are needed to improve adequately the encountered limitation to comply with regulatory and quality requirements. In this direction, recent data combining chemical analysis and bioassays tools using high performance thin-layer chromatography (HPTLC) coupled to genotoxicity testing look to be promising tools [18,19,20].

As a result of the presence of endocrine activity leaching from packaging material due to the presence of compounds such as Bisphenol A (BPA), further actions were undertaken to exclude endocrine-disrupting compounds from packaging material. Therefore, application of bioassays to assess endocrine activity potentially migrating from packaging material could play a role. This approach has been applied as a general screening tool captured in a study using a polystyrene-based plastic FCM anticipated to release 4-nonylphenol, an endocrine-active substance [21]. Analytical analyses of the migration sample confirmed the presence of 4-nonylphenol, and the bioassays detected endocrine activity which supported the viability of the approach. 

In 2019, the European Council of the Paint, Printing Ink and Artists’ Colours Industry (CEPE) laid out a safety assessment guideline specifically for metal can coatings similar to the complex mixture safety assessment strategy (CoMSAS) for carton food contact materials [22]. This safety assessment included guidance on all steps of CoMSAS such as the preparation of the metal can migration sample, analytical analysis of the migrants, genotoxicity evaluation and a risk assessment utilizing the TTC approach. 

A review of analytical techniques used for screening and quantification of unknown NIAS in FCMs was conducted by Nerin et al. [1] due to the lack of standardized methodology. The authors concluded that analytical results may differ due to the use of different methods and equipment. Differences in the preparation of the migration sample were also identified as a source of differing analytical results. The authors concluded that a comprehensive analysis of all migrants from FCMs is not possible with current knowledge and analytical equipment. As such, these authors reinforced the need for bioassays to assess the toxicity of the analytically undetermined migrants to manage risk. 

In alignment with published European regulatory guidance [3,5,15,23] and recent proposals for the safety assessment of NIAS [10,16,21,22,24], a detailed protocol for the chemical and biological screening of migrants from coated metal packaging materials during the product research and development (R&D) stage is proposed. The goal of this study is to apply the protocol, using can coatings as a case study, for the safety screening of new food packaging. The protocol focused on combining data from analytics and validated bioassays (i.e., endocrine activity, cytotoxicity and genotoxicity) for the risk assessment of can coating materials. This approach aligns with the ILSI “Guidance document on best practices on the risk assessment of NIAS in FCMs” [22] where the role of biological assays, chemical analysis and risk assessment are used to evaluate the safety of NIAS. 

The purpose of this study is to evaluate the protocol and guidance on a food contact coating with an inter-laboratory evaluation to determine the validity of the methods and reproducibility, including sample preparation, bioassays and chemical analyses. The study was conducted with the collaboration of a global food manufacturer, a global coatings manufacturer and three contract research laboratories with expertise in migration sample preparation, analytical chemistry and bioassays evaluating specific toxicological endpoints of concern.

## 2. Materials and Methods

### 2.1. Packaging Material

Metal can-based panels manufactured for the study were selected for the packaging material case study. The coating applied onto the metal can-based panels in this study is an experimental coating not intended for commercial use for food contact and was chosen solely for this study. Inclusion of a standard epoxy-based R&D formula comprised of Epon 1007 F epoxy resin (Hexion, Columbus, OH, USA) crosslinked with Cymel 227-8 urea crosslinker (Allnex, Frankfurt, Germany) was used as the reference “positive control” (PC), expected to trigger endocrine activity and genotoxic effect according to preliminary chemical analysis and biological activity characterization. After pre-validation confirmation of the selected “positive control” material (data not shown), the metal can panels were prepared by a packaging supplier including (1) coated panels (CT), (2) uncoated panels (UCT) and (3) epoxy-based (PC) panels. 

### 2.2. Inter-Laboratory Study Design

The crossover inter-laboratory study was designed as shown in Figure 1. The five laboratories participating in the study were coded LAB A, LAB B, LAB C, LAB D and LAB E. The criteria for the laboratories’ selection were based on certified expertise to perform the chemical analysis and/or the bioassays. The laboratories were designated to (1) prepare the migration simulation, (2) perform the chemical analysis, (3) perform the bioassays and (4) to collect chemical and bioassays data. Three laboratories were able to perform sample preparation and chemical analysis and three laboratories were able to perform bioassays. Only one laboratory was able to perform both types of analyses. Metal panels were coded and were blindly distributed to the designated laboratories. Three laboratories (LAB A, B and C) were requested to prepare migration simulation in triplicates for each R&D metal can coated sample, uncoated, epoxy-based and blank solvent process control. Three laboratories (LAB A, C and E) were requested to perform chemical analysis and three laboratories (LAB B, C and D) were requested to perform the selected bioassays.

The analysis of substances of very high concern (SVHC) or safety assessment of the packaging material used are out of the scope of the study. The analysis expects to (1) assess the reproducibility of the protocol and guidance and to identify the limitations and (2) evaluate the consistency of the chemical analysis and the bioassays between the laboratories. Specifically for the epoxy sample (PC), the induction of endocrine activity is expected as an endpoint for the bioassays.

### 2.3. Sample Preparation

Migrations of flat metal packaging samples and blanks were performed using commercial migration cells (Siegwerk migration cell system Sieg-Mi-Flex, LABC-Labortechnik, Bodensee, Germany) with a total exposure area of 1 dm^2^ (2 single-sided panels of 0.5 dm^2^ each) and 100 mL of the food simulant (50% ethanol). 

Migration experiments were performed using the food simulant and time/temperature conditions specified in the European regulation on plastics (No. EC 10/2011) [5] for dairy foods, i.e., 10 days at 60 °C in 50% (*v*/*v*) ethanol in triplicate. The simulant was removed from the migration cells and processed as described below [within Liquid–Liquid Extraction (LLE) section]. For preparation of the migration simulants, high purity chemicals for residue analysis were selected. Ethanol ≥99.8% for residue analysis, reference number 02851, was purchased from Sigma-Aldrich and dichloromethane (DCM) GC-MS SupraSolv^®^, reference 100668, used for solvent extraction was purchased from Merck. LC-MS grade water from a Millipore purification system was used. Quality of solvents used was assessed before migration testing to confirm absence of biological activity such as genotoxicity and endocrine activity to exclude any artifact from the results originated from the solvents used. 

### 2.4. Chemical Analysis

LAB A, C and E were requested to perform chemical analysis of the migration samples prepared by LAB A, B and C. Chemical analysis was focused on liquid chromatography–high-resolution mass spectrometry (LC-HRMS) for each sample tested in each lab. 

### 2.5. Liquid–Liquid Extraction (LLE)

LLE of the migration samples was used to facilitate the sample concentration and to avoid having water in the concentrate, which would complicate the GC analysis. The LLE was performed by transferring each food simulant migration sample individually to a separate funnel and extracting three times with 100 mL aliquots of DCM. The combined organic layers were then dried over sodium sulfate and filtered through a sintered glass funnel, or a funnel packed with glass wool (previously cleaned by heating at 400 °C overnight). The resulting dried organic solution was concentrated to a final volume of 1.0 mL using a sample concentration system available to the lab (LAB C used Rocket evaporator (Thermo Fisher Scientific, Waltham, MA, USA), LAB A and LAB E used a rotary evaporator (Büchi, Flawil, Switzerland)). To facilitate this step and accurately reach the final volume, evaporation under nitrogen flow was used when the final volume was close to 1.0 mL. If needed, the final volume was adjusted to 1.0 mL with DCM.

The same procedure was followed for the blank sample using 50% ethanol in a glass bottle with a ground glass stopper. These samples were labelled “solvent blank” to distinguish them from the panel blanks. All the concentrated samples were collected in glass vials using polytetrafluoroethylene (PTFE) caps and divided into three aliquots of 100 μL and one of 700 μL for the analytical (GC-MS, gas chromatography–flame ionization detection (GC-FID), LC-HRMS) and biological investigations, respectively. For the latter, 700 μL DMSO was added to the 700 μL DCM extract and the DCM was evaporated under nitrogen flow. Only small injection volumes of DCM were used for LC-HRMS to control for incompatibilities with C18. 

### 2.6. Internal Standards (IS)

D10-benzophenone (CAS 22583-75-1, Sigma-Aldrich, Buchs, Switzerland) was used as IS for the chemical analysis. The IS was spiked after migration, but before concentration, so that the final concentration of the IS in the samples was 16 ppb (16 µg/L) (corresponding to 10 ppb in food or 10 µg/6 dm^2^). 

### 2.7. Chemical Analysis Using LC-HRMS

#### 2.7.1. LAB A

An ultra-performance liquid chromatography (UPLC) system (I-Class, Waters^TM^, Milford, MA, USA) coupled to a Synapt G2-S mass spectrometer (Waters ^TM^, Milford, MA, USA) was used. The mass spectrometer was fitted with a heated ESI source (Waters^TM^, Milford, MA, USA). An ACQUITY BEH C18 analytical column (100 mm × 2.1 mm, 1.7 μm) (Waters^TM^) kept at 40 °C was used. The flow rate was set to 0.5 mL/min. Both mobile phases were composed of 0.1% formic acid (LC-MS LiChropur, Merck, Schaffhausen, Switzerland) in water (Mili-Q, Merck) (A) and acetonitrile (LiChrosolv, Merck) (B) and the gradient was as follows: 0–3 min, 95% (A); 3–6 min, 40% (A); 6–15 min, 5% (A). The positive-ionization mode was operated as follows: cone gas flow 30 L/h, desolvation gas 950 L/h at 550 °C, nebulizer 6 bar, sampling cone 30 V, source temperature 120 °C, capillary 0.5 kV. Positive MS data were recorded in full-scan resolution mode (*m*/*z* 50–1200) with 23,000 FWHM at *m*/*z* 556.

#### 2.7.2. LAB C

An ultra-high performance liquid chromatography (UHPLC) system (TLX-2) with Allegro quaternary pumps coupled to an Orbitrap Q-Exactive mass spectrometer (Thermo Fisher Scientific, San José, CA, USA) was used. The mass spectrometer was fitted with a heated ESI source (HESI, Thermo Fisher Scientific, San José, CA, USA). An ACQUITY BEH C8 analytical column (100 mm × 2.1 mm, 1.7 μm) (Waters Corporation, Milford, MA, USA) kept at 40 °C was used. The flow rate was set to 0.4 mL/min. Both mobile phases were composed of 0.5 mM ammonium acetate and 0.1% formic acid (grade) in water (A) and methanol (B) and the gradient was as follows: 0–0.5 min, 5% (B); 2.5 min, 35% (B); 10 min, 100% (B); 10–18 min, 100% (B); 18.5 min, 5% (B); 18.5–22 min, 5% (B). The positive and negative ionization switching mode was operated with parameters as follows: sheath gas flow 15 arbitrary units (AU); auxiliary gas flow, 5 AU; sweep gas flow, 1 AU; capillary temperature, 250 °C; heater temperature, 100 °C; spray voltage, +3500 kV and −2500 kV for the positive and negative mode, respectively; S-lens radio frequency, 70 AU. Positive and negative HRMS data were acquired simultaneously in full-scan (FS) and variable data independent acquisition (vDIA) mode. For data interpretation, only the positive ion mode was used. A resolving power full width half minimum (FWHM) were used at 35,000@200 and 17,500@200, for FS and vDIA mode, respectively.

#### 2.7.3. LAB E

LC-MS analyses were performed on a Waters (Milford, MA) ACQUITY UPLC H-Class with a quaternary solvent manager coupled to a Waters G2-XS QToF mass spectrometer. Analytical separations were performed by injecting 5 µL of each sample on an ACQUITY UPLC BEH C18 column (75 mm × 2.1 mm, 1.7 μm) with Mobile Phase B (0.01% formic acid in acetonitrile ramped linearly at a flow rate 0.4 mL/min from 30–98% (13 min with a 4 min hold) against Mobile Phase A (0.01% formic acid/0.03% NH_4_OH in water). Prior to analysis, the mass spectrometer was tuned and calibrated according to the manufacturer’s specifications. Automatic mass correction was performed for each run using leucine enkephalin for the positive and negative lock masses (lock scan performed every 40 s for ~1 s). Positive and negative mode continuum mass spectra were recorded separately in sensitivity mode from 170 to 2200 Da using ESI. The mass spectrometer source settings were as follows: capillary voltage, 3.0 kV; sampling cone voltage, 50 V; source temperature, 121 °C; source offset, 80 V; desolation temperature, 400 °C; cone gas flow, 50 L/h; and desolvation flow, 500 L/h. The quadrupole sector settings were as follows: LM resolution, 4.7; HM resolution, 15.0; aperture 1.0; pre-filter, 2.0; and an ion energy, 0.2; MS and MS/MS spectra were recorded simultaneously in MSE mode at 0.5 sec/scan at low collision energy (10 eV), and with a high collision energy ramp of 10–50 eV.

### 2.8. Chemical Analysis Data

For the intra-laboratory evaluation, substances were defined as consistent when the difference between their exact masses (-cation) and their retention times were lower than 0.05 Da or 0.05 min, respectively. The consistency was first assessed between the replicates for each sample and each laboratory. The number of substances having at least 2 consistent replicates was calculated. This analysis was conducted to assess the ability of each analysis laboratory to give repeatable results. Then, the results obtained by each laboratory for each extract (sample) were compared. For this comparison, only the substances with at least 2 consistent replicates were used. This analysis aimed at assessing the impact of the extraction on the global consistency.

Finally, for the inter-laboratory comparison, the exact mass of each substance was matched to all the potential exact masses using still the threshold of 0.05 Da. 

### 2.9. Inter-Laboratory Chemical Analysis Reporting

Each laboratory was requested to report:The details on sample preparation, such as the solvent supplier and migration conditions (time, temperature, food contact surface, volume of simulant);The color of the simulant at the end of the migration;Any precipitation or presence of particles during the sample prep;The details on the equipment used during the sample preparation (e.g., migration cells, equipment for concentration);Any deviation from the original extraction/concentration protocol. The instrument methods were chosen by the individual labs in this study, based on each laboratory’s currently employed methods;The details on LC and GC instruments (brand, type, etc.), methods and settings used for the analysis, as well as the chromatography conditions (column, gradient, standards).

### 2.10. Bioassay Analysis

LAB B, C and D were requested to perform bioassays of the migration samples prepared by LAB A, B and C. Effect-based analysis was performed to target potential effect on endocrine activity (nuclear receptor (NR) binding activities) and DNA damage (genotoxicity and mutagenicity assays).

### 2.11. Reagents for Bioassays

Dimethyl sulfoxide (DMSO), flutamide (CAS No. 13311-84-7), tamoxifen (CAS No.10540-29-1), menadione (CAS No. 58-27-15, purity > 98%), 17β-estradiol (CAS No. 50-28-2), 5α-dihydrotestosterone (CAS No. 521-18-6), benzo[a]pyrene (CAS No. 50-32-8), 4-nitroquinoline-n-oxide (CAS No. 56-57-5), cyclophosphamide (CAS No. 6055-19-2), 2-aminoanthracence (CAS No. 613-13-8), sodium azide (CAS No. 26628-22-8), 2-nitrofluorene (607-57-8) were purchased from Sigma-Aldrich (Buchs, Switzerland). Tributyltin acetate (CAS 56-36-0) was purchased by Merk (Darmstadt, Germany). Reference compounds concentration series in DMSO were purchased from BioDetection Systems Inc (BDS) (Amsterdam, Netherlands) (Amsterdam, the Netherlands) and the RealTime-Glo™ MT Cell Viability Assay from Promega (No. G9713) (Dübendorf, Switzerland). BlueScreen HC reagent materials were provided by Gentronix Limited (Alderley Park, UK).

### 2.12. Nuclear Receptor (NR) Binding Activities

NR-binding activation/inhibition effects were evaluated applying a panel of quantitative Chemical Activated Luciferase gene eXpression (CALUX) gene assays developed by BDS (Amsterdam, the Netherlands [25,26]. Licensed reporter cell lines were used: the Estrogen Receptor alpha (ERα) and the Androgen Receptor (AR) with the respective antagonistic version (Anti-ERα and Anti-AR) and the Aryl hydrocarbon Receptor (AhR). The tests were performed by LAB D and LAB C, adapting the OECD No. 455 [27] as well as OECD No. 458 for AR CALUX [28]. 

Briefly, cells were seeded in a 96-well plate (VWR, Dietikon, Switzerland) and incubated for 24 h at 37 °C, 5% CO_2_ and 100% humidity. After the incubation, cells were exposed in triplicates to concentration series representing a full dose–response curve of reference compounds for each test: 17β-Estradiol (E2) for ERα CALUX, Dihydrotestosterone (DHT) for AR CALUX and Benzo(a)pyrene (B(a)P) for AhR CALUX. In addition, a serial dilution of each of the different migrates were exposed in triplicates. After 24 h of exposure for cytotoxicity, ERα, AR and 4 h of exposure for poly-aromatic hydrocarbons (PAH) of exposure, the luciferase activity is measured with a luminometer plate reader (Mithras, Berthold, Germany).

Evaluation of antagonistic effect was performed at fixed concentration (EC_50_) of agonist in presence of serial dilutions of the antagonistic reference. For the Anti-AR CALUX, the agonist is DHT at the EC_50_ 0.3 nM and the antagonist is Flutamide. For the Anti-ERα CALUX, the agonist is E2 at the EC_50_ 6.0 nM and the antagonist is Tamoxifen. Blank, negative control and cell control were tested on each plate.

To discriminate between activity induction and potential cytotoxic effect, two approaches were applied: (LAB C) used an in-well multiplex method RealTime-Glo™ MT Cell Viability Assay (Promega, Dübendorf, Switzerland) [29] and (LAB D) used a cell line designed for that purpose (Cytotox CALUX) using the tributyltin acetate substance as reference [30]. 

### 2.13. Genotoxicity and Mutagenicity Assays

#### BlueScreen™ HC Test

The Bluescreen™ HC test (Gentronix Limited, UK) uses the human-derived, p53-competent, TK6 cell line stably transfected to express the GADD45a-Gaussia luciferase construct (Gluc cells) [31]. After exposure of this reporter cell line to genotoxic substances, the GADD45a gene is upregulated, leading to an accumulation of Gaussia luciferase that can be quantified to determine genotoxicity. This signal is corrected against cell number measured using thiazole orange fluorescence, giving a relative luminescence unit (RLU). The thiazole orange measurement also gives an estimate of cytotoxicity. The overall methodology used was as outlined by [31] and method supplier Gentronix Limited, with amendments to the metabolic activation conditions as detailed below. Briefly, dose–response analyses of prepared migrates were performed using the recommended protocol for BlueScreen™ HC assay, with GLuc cells treated in duplicates with each concentration of the prepared migrates. Testing was conducted in both the absence and presence of rat liver S9 fraction and cofactor mix, using BlueScreen™ HC reagent kits (Gentronix Limited, UK). For studies conducted in the presence of S9, cells were exposed to 0.125% (Aroclor 1254-induced male SD rat liver (Moltox)) for 16 h. Following the S9 exposure period, S9 mix was then removed by centrifugation, cells washed and then incubated in recovery medium (Gentronix Limited, UK) for a further 32 h. After incubation, GADD45a-Gluc luminescence reagents were injected with appropriate buffer and the relative luminescence units (RLU) were recorded. Cytotoxicity was assessed by addition of a cell lysis reagent and thiazole orange labelling and relative fluorescence units (RFU) were collected with a plate reader (ex485 nm/em535 nm). 

### 2.14. Bacterial Reverse Mutation Test (MiniAmes Test) 

In addition to the inter-laboratory bioassays testing, the MiniAmes bacterial mutation assay was included in the study to exclude mutagenicity potential. The test was performed by LAB B, adapting the OECD guideline 471 [17] plate-incorporation method, using only migration samples produced at their facilities. As a screening test, two *Salmonella typhimurium* histidine auxotrophic strains (TA98, TA100) (Gentronix Limited, UK) were used in the absence and presence of Aroclor-1254 induced rat S9 liver fraction (Moltox), with plating conducted on 24-well minimal agar microplate. The R&D coated panel selected at the outset of the study was chosen for activity within the CALUX and BlueScreen™ HC endpoints, but not the MiniAmes as this endpoint was added following initiation of the project. To enable confirmation that known mutagenic substances can be detected following an extraction and concentration procedure, a simulated B(a)P was produced. A volume of 50% ethanol was prepared with a concentration of 0.5 µg/mL B(a)P, concentrated using the same procedures as those undertaken for the coated, uncoated and procedural blank, returning a concentrated sample at approximately 50 µg/mL in DMSO, which was used in the Ames test as a procedural control for detection of mutagenic substances, at an assumed maximum on plate dose of 0.25 µg per plate (5 µL of DMSO stock added per well).

The TA98 and TA100 strains were exposed to a half-log serial dilution of prepared migrates including the simulated B(a)P extraction, with triplicates wells used at each tested concentration per strain per S9 condition. In addition, strains were exposed to the vehicle (DMSO) control and standard positive controls (2-aminoanthracene, 2-nitrofluorene, sodium azide). The 24-well agar plates were then incubated at 37 °C for 3 days. Following the incubation period, the surface of each microplate well was analyzed visually to determine the number of revertant colonies, as well as an estimation of background lawn and any visible precipitate. 

### 2.15. Biological Testing Data Analysis

#### 2.15.1. NR-Binding Activities

Relative luminescence units obtained were processed as described by proprietary calculation template (BDS) (Amsterdam, The Netherlands). Receptor-binding activation was quantified as the amount of relative luciferase units (RLU) emitted after samples’ exposure. The signal is proportional to the amount of ligand-specific receptor-binding activation. After normalization to solvent control (DMSO) the percentage of relative induction (RI) or relative inhibition (RInh) was calculated for the selected nuclear receptors and the cell viability. The samples were then expressed as percentage of the maximum reference compound response. All dose responses and dilutions were performed in technical triplicates. The threshold for cell viability was below 80% according to method supplier recommendations. The threshold for positive reporter gene activities for agonistic activities in ERα and AR CALUX was above 5% and antagonistic activities for ERα and AR CALUX was below 80% according to method supplier. For the PAH CALUX bioassay, the threshold level was set as 10%. Reference compounds, negative control (Blank) and cytotoxicity (menadione or tributyltin acetate) were processed on each plate to confirm the quality and validity of tests. Dose–response curves were plotted using GraphPad Prism (version 9.02 for windows) for agonistic or antagonistic modes. The effective concentration (EC_50_), the inhibitory concentration (IC_50_), the corresponding equivalents (EQ) in reference compound for each CALUX test (e.g., estradiol equivalents (EEQ)) per dm^2^, the limit of detection (LOD) and the limit of quantification (LOQ) were calculated using the proprietary BDS proprietary quantification template version 3 (BDS). Dose–response obtained between laboratories C and D was compared. 

#### 2.15.2. BlueScreen™ HC Test 

The luminescence was normalized to the fluorescence signal to correct for variation in cell yield caused by cytotoxicity. The final signal was directly proportional to the induction of GADD45a and/or the cytotoxicity. Blank, negative control (cells + solvent) and positive control (cells + mutagenic substance) was tested in each plate. For the test performed without S9, the genotoxicity threshold is set at a relative luminescence induction of 1.8, i.e., an 80% increase over the constitutive expression of GLuc. In presence of S9, the genotoxicity threshold is set at a relative luminescence induction of 1.5. A result is positive if the fold induction respect vehicle control is ≥1.5 the threshold defined as the lowest effective concentration (LEC). Cytotoxicity results are scaled such that the average final cell density of the vehicle-treated controls (using the fluorescence intensity measurements) equals 100%. The cytotoxicity threshold is set to 80% of the relative cell density of the vehicle-treated control for both conditions.

#### 2.15.3. MiniAmes Test 

The number of revertant colonies determined at each dose was then compared to the spontaneous revertant rate in the solvent control and a fold increase determined. A response was considered positive for bacterial mutagenicity when a ≥2-fold increase in the mean number of revertant colonies over the respective solvent control mean value for strains TA98 and TA100 was met, and there was also evidence of a clear dose-dependent response, in either the absence or presence of metabolic activation (±S9).

### 2.16. Inter-Laboratory Bioassay Reporting

Each laboratory was requested to report:Details on the sample exposure to cell lines (e.g., cell passage, exposure time);96-well plates, white or transparent;Duration of experiments;Deviation from shared protocol;Use of DMSO batch provided for all laboratories;Indicate technicians performing test (for technical reproducibility issues);Quality criteria for each experiment (e.g., reference controls, LOD, LOQ, LEC, cell passages, test performance).

## 3. Results

### 3.1. Chemical Analysis

The initial study was designed to include both GC and LC analyses. As one of the three participating labs did not provide information on the fragments observed, the GC results could not be properly compared and, therefore, were not considered further. This section focusses on LC-HRMS only.

#### 3.1.1. Intra-Laboratory Variation by LC-HRMS

The intra-laboratory variation of the triplicate measurements for each of the participating labs was investigated. To assess this variation, an arbitrary selected range of the 12 most intense substances detected in each extract were reported by each lab. The assessment was performed on both the positive control (PC) and coated (CT) samples in triplicate to understand whether the intra-laboratory variation was consistent between different samples. 

The “analysis lab” reported in Table 1 was the laboratory that performed the LC-HRMS analysis. The laboratory that performed the preparation of the extracts to be analyzed (by the analysis lab) is reported in the “sample” column. The number of substances present in at least two replicates highlights how well the three laboratories were aligned. This is reported in Table 1 as (n/N), where n equals the number of accurate neutral masses that were observed in at least two of the three intra-lab replicates for each individual lab, and N equals the total number of unique neutral accurate masses across the three replicates for each individual lab.

For samples that contained multiple identical masses (within Δ 0.05 Da) at different retention times corresponding to isomers, only one of the multiplicate masses was counted in calculating n/N. Reporting only one isomer was adopted since isomers across the labs were not identified and, thus, could not be conclusively determined to be paired or not. Under this reporting convention, 12/12 would mean perfect alignment between the three laboratories whereas 0/36 would mean no alignment between the three laboratories.

The results in Table 1 demonstrate that the triplicate measurements are fairly consistent. Similar consistency was observed for both the coated sample and the positive control.

The three laboratories received the same instructions to prepare extracts of the samples. To understand whether the procedure of preparing the extracts followed by the three different laboratories was consistent, one sample was chosen to be extracted by each laboratory and the extracts were shared with the other laboratories. One sample was considered sufficient to demonstrate this. These three extracts were then analyzed by the three laboratories by LC-HRMS to determine whether the laboratories were aligned in the sample preparation. Note that each chemical analysis laboratory received the same exact three extracts prepared by the other labs.

Table 2 demonstrates the LC-HRMS results. Note that this table reflects the two consistent replicates from Table 1. These values are indicated for the different laboratories in the extract column. For example, if the extraction results of two extracts were aligned, 12 should have been reported for the first line. In the case of total misalignment, 0 would have been reported. As example, for the first three lines of Table 1, one substance was not in common to the other extracts. Therefore, only 11 was reported in Table 2. 

The number of substances that were determined to be common across extracts prepared by two different laboratories was relatively high. There was only one outlier, for which no explanation was found.

#### 3.1.2. Inter-Laboratory Variation by LC-HRMS

To investigate the inter-laboratory variation, one sample, CT extracted by LAB C, was chosen for analysis by the three participating laboratories. 

Each of the three laboratories was requested to report an arbitrarily selected range of the 100 most intense substances that were observed in at least two of the three replicates of the sample (LAB C-CT). The mass and intensity of the cationized molecule was reported. Only the mass and intensity of the most intense ion was reported to avoid reporting a substance multiple times as different cation adducts of the same substance are typically observed.

Due to the different mass ranges of the instruments used or different ranges acquired, it was decided to narrow the mass window and focus only on ions with an *m*/*z* between 300 and 1000. This range was covered by all three labs, allowing for a standard range to compare. It should be noted though that those masses above and below this range were also collected by some labs. These broader ranges are likely to have little to no effect on the mass resolution and accuracy for the masses that were reported.

By modifying the reporting range to *m*/*z* 300–1000, the number of substances was reduced to 75 for LAB C, to 72 for LAB E and to 35 for LAB A (Figure 2). Of the 35 substances reported by LAB A, 20 compounds were reported by LAB C and LAB E as well. The consistency between LAB C and LAB A was the best, as 27 substances of the 35 reported by LAB A were in common. Though LAB C and LAB E reported the highest number of substances, the number of substances reported in common was relatively low (33 substances in common out of a maximum of 72).

The main factors that impacted the consistency in the reported results was: (1) The presence of isomeric structures making it difficult to determine if two labs reporting a mass were reporting the same compound; (2) The instrumental methodologies between the different labs were not required to be fully aligned to mimic realistic conditions used in each lab; (3) The study design dictated that each of the labs follow their own in-house methods; and (4) The detected substances were not identified but only specified by exact mass and retention time. Furthermore, the retention time was of little value in comparing results from different labs since different chromatography methods were employed.

### 3.2. Bioassays Analysis

The solvent control, the coated and uncoated panels developed for the inter-laboratory study were tested to assess biological activities using in vitro toxicology tools. Effect-based dose–response and quantitative bioanalysis targeting nuclear receptor and genotoxicity endpoints was performed. 

The nuclear receptor assessment was performed using the CALUX assays including the endocrine hormones to evaluate the effect on the estrogenic responsive ERα and androgenic responsive AR and were both tested for the agonist and antagonist mode, as well as a metabolism-related endpoint, the AhR in the PAH CALUX. The cell viability was evaluated using two different approaches (RealTime-Glo™ and Cytotox CALUX) as described in the method section. 

The genotoxicity assessment was performed using the BlueScreen™ HC test in the presence and absence of metabolic activation. In addition, evaluation of mutagenicity was also assessed as an extra analysis using the MiniAmes test (by LA B only), outside of the scope of the original inter-laboratory study but generated as complementary information. 

Quality control to monitor inter- and intra-laboratory method performance was assessed by evaluating the reference control dose–response curves and historical standard range values such as EC_50_, IC_50_, LOD, LOQ for CALUX assay and positive controls (LEC’s) for BlueScreen™ HC and MiniAmes tests. 

#### 3.2.1. NR-Binding Activities 

Performance of the CALUX assays was confirmed using the reference controls with dose–response curves obtained for each laboratory. The quality criteria for each test were achieved. 

Migration simulation samples prediluted in DMSO were tested by LAB C and D in a dose–response manner using the CALUX assays coding for the ERα and AR (agonistic and antagonistic modes) and AhR. Quality controls for all the endpoints tested were achieved.

##### Estrogen Receptor (ERα)

No dose–response effect was recorded indicating agonistic effect for the estrogen receptor with any of the migration simulation samples by LAB C or D (data not shown). An antagonistic effect towards estrogen receptor activation above the threshold (>20%) to consider the sample as antagonistic was recorded by both laboratories for the triplicates of the positive control migration samples (A, B and C) (Figure 3). The recorded relative inhibition and the corresponding equivalents in tamoxifen (μg) equivalents per dm^2^ were calculated (Table 3). No antagonistic effect was observed with the coated or uncoated panels or the solvent control (blank) sample.

##### Androgen Receptor (AR)

No dose–response effect was recorded indicating agonistic effect for the androgen receptor with the migration simulation samples by LAB C or D (data not shown). However, an androgen-antagonistic effect above the threshold (>20%) to consider the sample as antagonistic was recorded by both laboratories for the triplicates of the positive control migration samples (A, B and C) (Figure 4). The recorded relative inhibition and the corresponding equivalents in flutamide (μg) equivalents per dm^2^ were calculated for all samples (Table 4).

##### Aryl Hydrocarbon Receptor (AhR) (PAH CALUX)

A dose–response effect was recorded above the 10% threshold to consider the sample as inducer of the AhR with stronger induction observed by PC samples except for PC sample tested by LAB C. In general, relatively high standard deviation range was observed for samples tested by both laboratories. Samples of LAB A and C displayed some discrepancies with sample CT (Figure 5). The recorded relative induction and the corresponding equivalents in B(a)P (ng) equivalents per dm^2^ were calculated for all samples (Table 5).

Finally, the quality criteria recommended either by the OECD guidelines or historical values of method developers were considered to confirm the performance of the tests in each laboratory participating to the study. The dose–response curves for the reference controls tested during the study for Anti-AR, Anti-ERα and AhR CALUX in LAB C and D are shown in Figure 6 demonstrating the consistency between laboratories. In addition, the standard ranges for reference controls tested are shown in Table 6.

According to the generated data in the current the inter-laboratory study performed, both LAB C and D were at the recommended ranges, confirming the quality of the tests performed.

#### 3.2.2. Genotoxicity 

##### BlueScreen™ HC

Following sample preparation by LAB A, B and C, concentrated migration samples were then analyzed in the Gadd45α-Gluc reporter screening assay, in both LAB B and LAB C sites. Both laboratories had previously demonstrated proficiency in the conduct of the BlueScreen™ HC assay procedure in the methodology first described by [31] and in the modified S9 conditions initially developed by LAB C to improve the limit of detection of the test system for genotoxic substances requiring S9 metabolic activation.

The means of the dose–response profiles of the replicate samples of the positive control panel and R&D coated panel from each preparation, analyzed by both LAB B and LAB C, are shown in the absence (Figure 7) or presence (Figure 8) of S9. BlueScreen™ HC genotoxicity data for both conditions are summarized in Table 7.

For the uncoated panel and procedure blank, both LAB B and LAB C BlueScreen™ HC results were consistent with a clear negative result in the assay, both with and without S9 metabolic activation across all samples analyzed from the three preparation lab sites (dose–response data not shown). For the coated panel, no significant induction of the Gadd45α-GLuc reporter was observed for any of the LAB A, B or C samples in both analysis labs in the absence of S9 activation. Cytotoxicity was observed at the higher concentration for these samples in both laboratories, which was consistently of higher severity in LAB B analyses than LAB C. Both labs produced data that met all assay acceptance criteria. In the presence of S9, the coated panel produced a positive result for genotoxicity in LAB B-prepared samples, only when analyzed by LAB B. The same sample analyzed at LAB C produced a sub-threshold result and was considered negative. It is important to note that LAB B prepared and analyzed samples that were not subject to transit procedures and the resulting delays between preparation and analysis. Cytotoxicity profiles were also observed for coated panels assessed in the presence of S9, though these were lower in severity and for LAB C analysis were less consistent for LAB A and LAB C preparations to those observed by LAB B for the same samples.

In the absence of S9, the positive control panel produced a genotoxic result in the LAB B analysis for samples prepared in LAB B and C. Whilst the LAB A samples were just below the 1.8-fold positive response threshold in the mean data, these did produce a positive result in one of the three replicates analyzed at LAB B. For LAB C analyses, all three sets of samples preparations were negative for genotoxicity, though the cytotoxicity profiles were of reduced severity as well. In the presence of S9, the positive control panel induced a genotoxic result at both LAB B and LAB C for samples prepared at LAB B and LAB C. LAB A prepared samples produced an above-threshold positive genotoxicity result for LAB B-analyzed samples, but not in LAB C analyses, though two of the three replicates did achieve at-threshold responses (1.46-fold and 1.42-fold increases) at the maximum dose analyzed for LAB A preparations in LAB C. 

For the positive control (PC) sample, a reporter induction was observed in the absence and presence of metabolic activation by both LAB B and C except for the sample prepared by LAB A. Considering the acceptance >1.5 threshold criteria to consider reporter induction as positive, only the ones highlighted in grey in Table 7 can be considered positive for Gadd45α induction, therefore confirming the lack of concordance for the final call for these samples. 

Overall, for the uncoated panel and procedure blank, a high degree of consistency was achieved between all preparations analyzed in both laboratories. For the positive control panels, genotoxicity signatures were detectable in some but not all the sample preparations. In addition to the variability of the sample preparation process, one of the possible parameters responsible of this inconsistency was the cell passage. Indeed, for one of the labs, the cell passage was close to the maximum passage allowed for the test.

##### Mutagenicity–Mini-Ames Test

Following the establishment of the initial study design, in which assessment of genotoxicity was to be conducted using the Gadd45α-Gluc BlueScreen™ HC reporter system, inclusion of the assessment of potential DNA-reactive migration products was considered of interest. To accommodate this additional assessment, LAB B conducted analyses of samples prepared within its own laboratory (LAB B preparations) using a 24-well plate, agar-based miniaturized Ames test with TA98 and TA100 strains with and without S9. As the positive control panel had been selected based on nuclear receptor and BlueScreen™ HC activity and not DNA-reactive mutagenicity, an alternative positive control for the process was incorporated into the mutagenicity testing. B(a)P was used as a simulated migration product, prepared in 50% ethanol, concentrated using the same procedures as all other samples and then tested in the 24-well Ames test alongside the R&D coated panel, uncoated panel, process blank (50% ethanol concentrated in same procedure) and concurrent Ames assay positive and vehicle (DMSO) controls. Testing was performed in triplicate, with each replicate sample tested on three microplate wells, per concentration across a total of eight doses. The results of this analysis are summarized in Table 8.

For both the R&D coated panel and uncoated panel, all samples analyzed produced a clear non-mutagenic profile in both TA98 and TA100 strains in the both the presence and absence of S9. The process blank was also clearly non-mutagenic in all test conditions, with concurrent Ames test positive controls generating data consistent with expected ranges and the vehicle control within historical control data in all tests. With regards to the B(a)P-simulated extracted samples, this produced an expected non-mutagenic result in both TA98 and TA100 strains in the absence of S9, given B(a)P requires CYP metabolism to produce its mutagenic metabolites. When tested in the presence of S9, B(a)P-simulated extraction produced a clear and dose-dependent mutagenic signal in both TA98 and TA100. These data indicate that potential presence of DNA-reactive mutagenic migration and extraction products is detectable using this overall bioassay approach.

## 4. Discussion

Following regulatory body recommendations to assess packaging safety by applying chemical methods and biological assays, the current inter-laboratory study was set up to evaluate the consistency of the data between different laboratories applying a protocol that would use those tools. For that, a coated and uncoated panel were prepared including an epoxy coating as a positive control. Five different laboratories participated in the inter-laboratory study. The reported chemical and biological analyses were collected to evaluate the intra- and inter-laboratory data consistency. The study aim was not to assess the safety of the material tested but to demonstrate the critical steps contributing to any inconsistency between laboratories that may lead to misleading and misinterpreting the safety evaluation of packaging material. 

There is a need for harmonization of testing protocols to reduce laboratory variability and to improve consistency. The protocol used for this inter-laboratory study incorporated several bioassay endpoints and chemical analyses for the evaluation of the migration samples from a coated metal panel. The protocol for the preparation of the migration samples was defined prior to starting the study by combining best practices used in the different analytical labs, with some steps/techniques being new to certain laboratories. It was noted that the triplicate analyses performed by one laboratory’s chemical analysis did not meet the expected quality, as a lack of reproducibility of the results was observed. There was, however, good correlation of the analytical results from the three different labs. Based on a recent article by [1] (published after this study was completed), the authors expressed concern that different analytical equipment may give different results and, thus, make meaningful comparison of the chemical analytical results from the laboratories in this study potentially very difficult. Despite this, the outcome of the chemical analytical results in this investigation was better than one would have expected based on the variety of analytical methods and equipment that were utilized for the study.

The laboratories evaluating endocrine activity and genotoxicity were selected following the recommendation of (1) EFSA scientific opinion [32] stating the importance of assessing genotoxicity potential for food contact materials and (2) regulators expressing increasing concern about endocrine disruption. The results from the bioassays showed good overall consistency between the laboratories conducting evaluations on the migration samples. Initially, appropriate data between the laboratories were observed for three reference substances with known endocrine activity. The solvent control (blank) and uncoated panels all tested negative for agonistic and antagonistic effects on estrogen and androgen endpoints (100% consistency). The consistency for the experimental coated panels (the unknown) and the epoxy-based coated panels (the positive control) was also good (75–100% depending on the endpoint). For example, both labs reported positive results for anti-ERα for the positive control panels, as would be expected for an epoxy coating. The experimental coating panels showed very consistent negative results for all endpoints (ERα, anti-ERα, AR and Anti-AR). For the genotoxicity evaluations, there was also high concordance (84%) between the two labs with migration samples showing a negative response. The consistency observed in this inter-laboratory study was clearly promising but learnings from this study suggest that further standardization and improvements are needed. We believe that an updated protocol would likely help to reduce the variability observed between the laboratories. The following observations that probably had the most impact on the quality of the data and ultimately resulted in discrepancies of the results are given below and suggestions for improvement of the protocol are made. 


MIGRATION SAMPLE PREPARATION CONSISTENCY. A likely cause of discrepancies in the bioassay and chemical analytical results is the migration sample preparation. The multi-step process that includes liquid–liquid extraction, drying, filtering, followed by concentration may result in different results for the samples (performed in triplicate) due to differences in the equipment used, technique and/or contamination of the sample. For example, the triplicates prepared by one lab did not result in the expected repeatability possibly due to contamination, the large number of manual steps involved and difficulties to dry the dichloromethane. Due to the negative environmental impact and safety concerns of halogenated solvents, such as DCM used here, it is recommended that an alternative solvent be identified to reduce the number of solvents for sample preparation required for testing. STANDARDIZED CHEMICAL LAB EQUIPMENT. Different laboratories made use of different analytical infrastructure that may have contributed to the variation in the results. For example, different evaporation systems and glassware were used between the labs. CHEMICAL ANALYTICAL EQUIPMENT METHODOLOGY. Different laboratories used different mass spectrometers and methods. Recommending a certain MS technology or brand cannot be achieved, as different laboratories use different mass spectrometers for various reasons. The current study highlights that different MS methods measure more efficiently in different *m*/*z* windows and the ionization efficiency for different compounds on different MS systems may be impacted as well. The LC system used was from different suppliers and solvents/gradients/columns used were different as well. Follow-up studies would be needed to better understand the impact of the MS technology, methods and LC conditions to improve consistency between results. SUBSTANCE IDENTIFICATION. The current study was designed such that identification of substances was not requested. This made the final interpretation of the study complicated as, for example, isomers could not be correlated between different labs (same *m*/*z*). Future inter-laboratory studies should request identification to be performed.STANDARDIZED BIOLOGICAL ASSAY PARAMETERS. The choice of cytotoxicity test and the resulting assessment for the endocrine evaluations differed for the labs, which sometimes yielded differences in the final call of the result (positive or negative). Guidance to conduct a cytotoxicity assessment and cell passage recommendations could lead to a greater harmonization on the call of the results. USE A HARMONIZED FOOD CONTACT PACKAGING SAFETY PROTOCOL. Finally, refinements of the current protocol should be made that incorporate the recommendations raised in this study. Some of the recommendations will also require additional research to determine their role in the observed variability prior to incorporating into the protocol. Part of defining the final protocol would be to apply the procedure in routine and testing the final protocol for consistency in another inter-laboratory study.


The R&D protocol for food contact coatings used in this inter-laboratory evaluation is an important step in identifying critical steps for best practices for safety assessment of food contact materials including coatings. The results from this study show that the methods were valid, and the results were generally reproducible. A revised protocol along with an additional qualification evaluation should be the next step in the path forward for an improved evaluation of the safety of packaging materials, such as can coatings. Harmonization between laboratories for packaging safety assessment of IASs and NIASs has been identified as crucial to reduce laboratory variability and to improve consistency. 

## Figures and Tables

**Figure 1 toxics-11-00156-f001:**
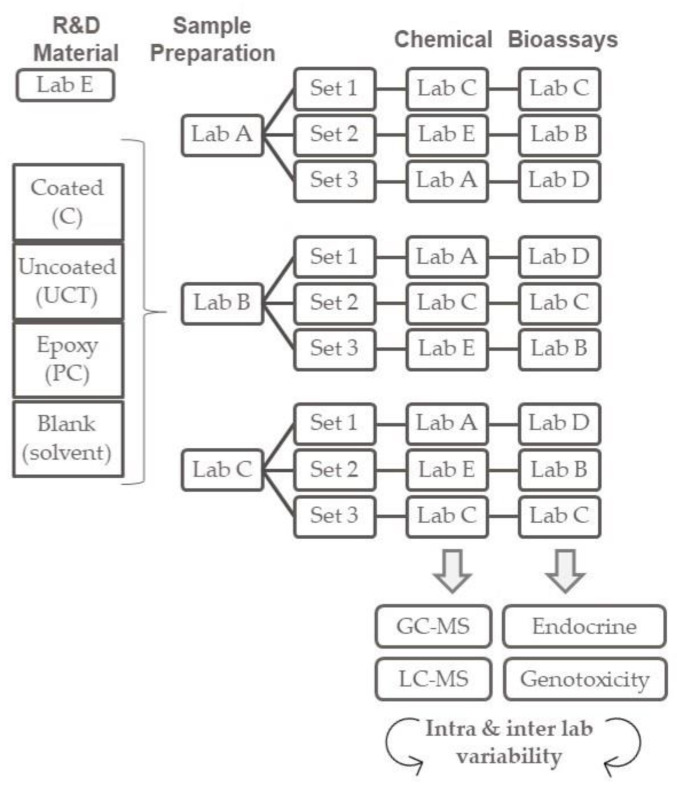
Crossover inter-laboratory study design to evaluate intra- and inter-laboratory variability of participants LAB A, LAB B, LAB C, LAB D and LAB E. Each set of samples consist of triplicates.

**Figure 2 toxics-11-00156-f002:**
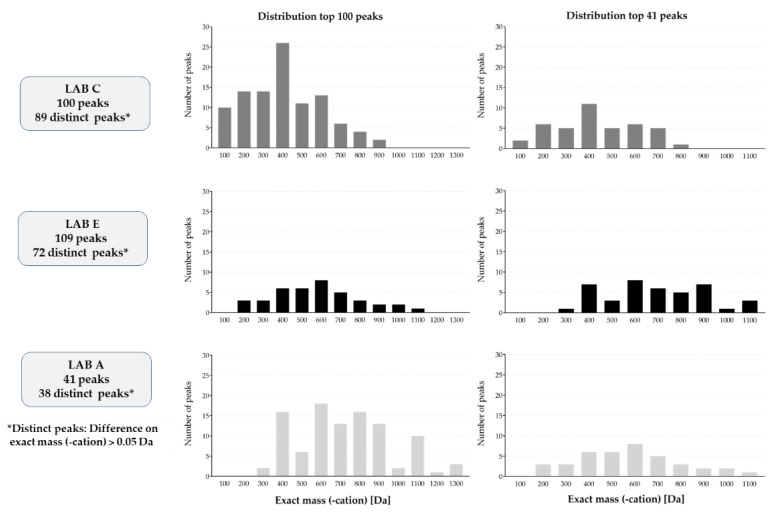
Inter-laboratory variation by LC-HRMS. Data collected from LAB C, E and A.

**Figure 3 toxics-11-00156-f003:**
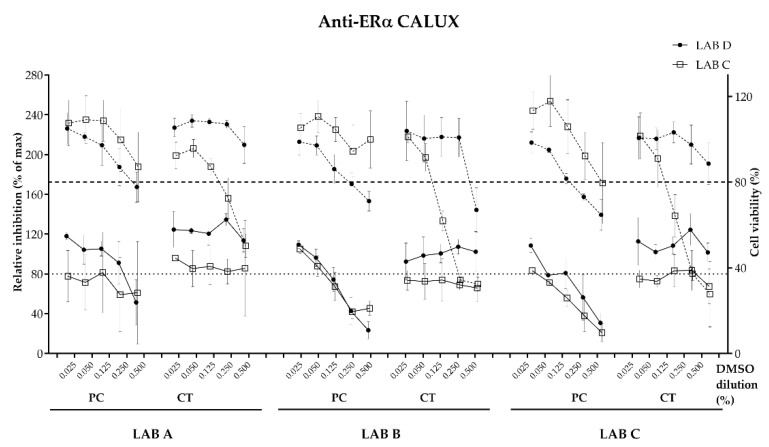
Dose–response curves with PC (positive control) and CT (coated metal panel) measured with the anti-ERα CALUX assay assessed by labs C and D on samples prepared in LAB A, B and C (DMSO dilution %). Graphs shows mean and standard deviation of each experimental point (3 biological replicates performed in 3 technical replicates per compound). Threshold relative induction at 80% (dotted line), below which is consider antagonistic. Threshold cell viability 80% (dashed line), below which is considered cytotoxic.

**Figure 4 toxics-11-00156-f004:**
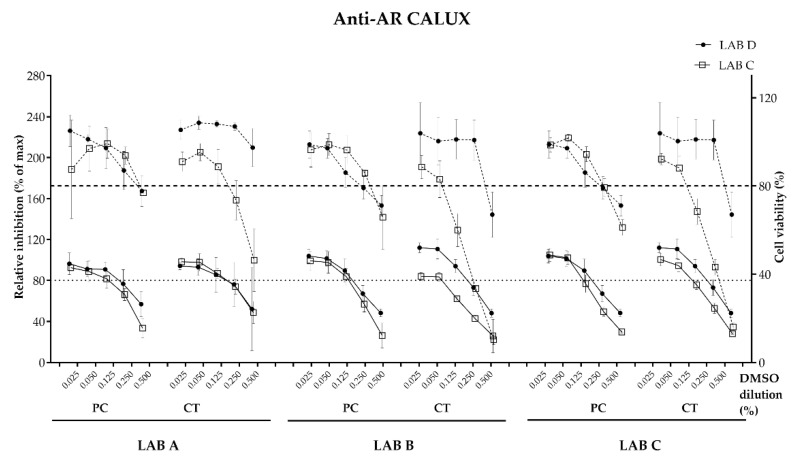
Dose–response curves with PC (positive control) and CT (R&D coated metal panel) measured with the anti-AR CALUX assay assessed by LABs C and D on samples prepared in lab A, B and C (DMSO dilution %). Graphs shows mean and standard deviation of each experimental point (3 biological replicates performed in 3 technical replicates per compound). Threshold relative induction at 80% (dotted line), below which is consider antagonistic. Threshold cell viability 80% (dashed line), below which is considered cytotoxic.

**Figure 5 toxics-11-00156-f005:**
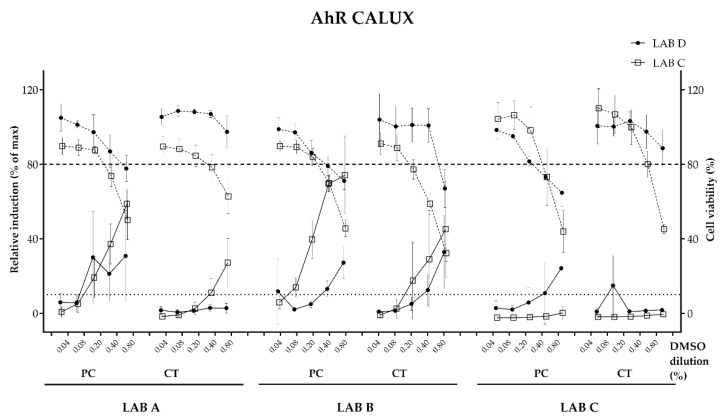
Dose–response curves with PC (positive control) and CT (R&D coated metal panel) measured with the AhR assay assessed by labs C and D on samples prepared in lab A, B and C. Graphs shows mean and standard deviation of each experimental point (3 biological replicates performed in 3 technical replicates per compound).

**Figure 6 toxics-11-00156-f006:**
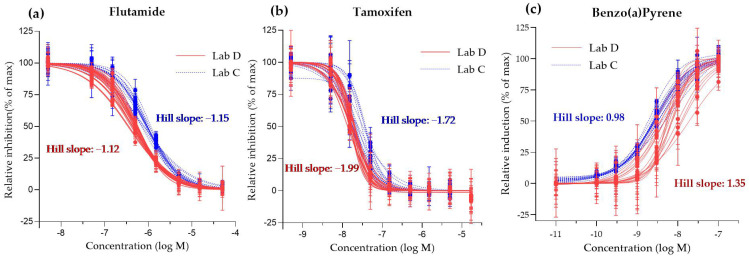
Dose–response curves of reference standards: (**a**) flutamide, (**b**) tamoxifen and (**c**) benzo(a)pyrene, respectively, measured with the Anti-AR, Anti-ERα and AhR CALUX assay assessed in LAB C and D. Graph shows mean and standard deviation of each experimental point (3 biological replicates performed in 3 technical replicates per compound).

**Figure 7 toxics-11-00156-f007:**
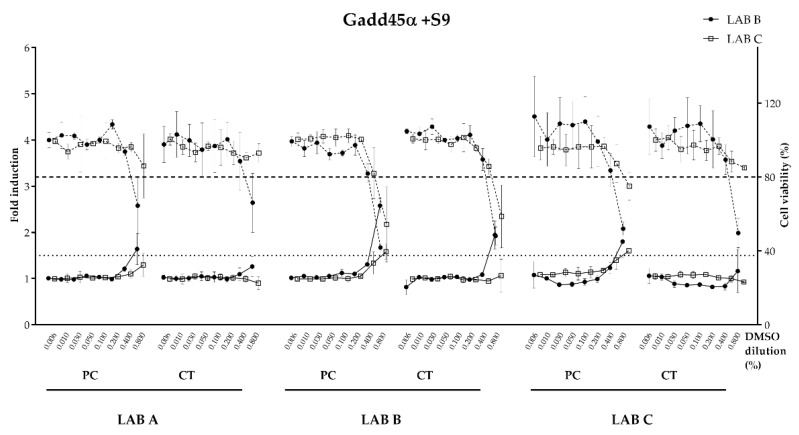
Dose–response curves with PC (positive control) and CT (coated metal panel) measured with the BlueScreen HC™ Gadd45α-Gluc assay conducted in LAB B and C on samples prepared in LAB A, B and C, in the absence of S9 metabolic activation (DMSO dilution in %). Relative luminescence fold induction (solid line with markers) and relative cell density (dashed line with markers). Threshold fold induction at 1.8 (dotted line), above which is consider genotoxic. Threshold cell viability 80% (dashed line), below which is considered cytotoxic.

**Figure 8 toxics-11-00156-f008:**
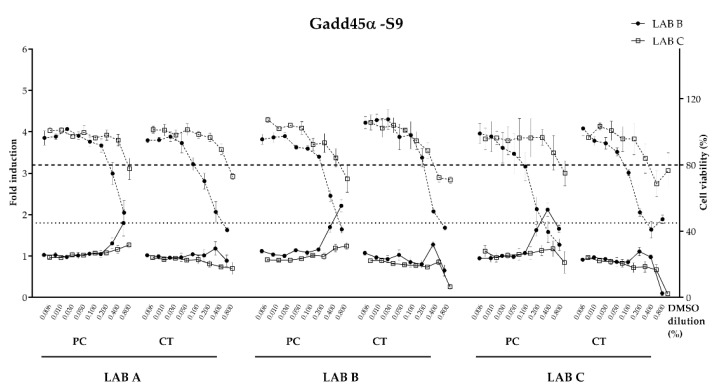
Dose–response curves with PC (positive control) and CT (coated metal panel) measured with the BlueScreen™ HC Gadd45a-Gluc assay conducted in LAB B and C on samples prepared in LAB A, B and C, in the presence of S9 metabolic activation (DMSO dilution in %). Relative luminescence fold induction (solid line with markers) and relative cell density (dashed line with markers). Threshold fold induction at 1.8 (dotted line), above which is consider genotoxic. Threshold cell viability 80% (dashed line), below which is considered cytotoxic.

**Table 1 toxics-11-00156-t001:** LC-HRMS analysis intra-laboratory consistency for CT samples.

Sample	Analysis Lab	Sample ^†^	Ratio n/N ^‡^
Coated	LAB C	LAB B	12/13
		LAB C	12/13
		LAB A	11/15
	LAB E	LAB B	11/17
		LAB C	12/19
		LAB A	10/19
	LAB A	LAB B	12/18
		LAB C	12/17
		LAB A	11/18

^†^ Triplicate extract prepared by ^‡^ Number of substances with at least 2 consistent replicates (n = masses observed in at least 2 samples, N = total number of unique masses).

**Table 2 toxics-11-00156-t002:** Consistency between laboratories for LC-HRMS analysis.

Sample	Analysis Laboratory	Extract	A ^†^	B ^‡^
Coated	LAB C	LAB A (11)	11	12
		LAB B (12)		
		LAB C (12)		
	LAB E	LAB A (10)	9	11
		LAB B (11)		
		LAB C (12)		
	LAB A	LAB A (11)	11	12
		LAB B (12)		
		LAB C (12)		

^†^ A: Number of substances common to 2 extracts ^‡^ B: Max number of substances in common to 2 extracts from Table 1.

**Table 3 toxics-11-00156-t003:** Data obtained by Lab D and C for Anti-ERα CALUX.

Assay	Data	Anti-ERα					
Sample		LAB A		LAB B		LAB C	
Assay		LAB D	LAB C	LAB D	LAB C	LAB D	LAB C
Positive control	RInb (%)	61.6 ± 5.5	NA	64.6 ± 8.1	62.0 ± 6.9	75.1 ± 7.8	58.7 ± 12.2
	TEQ	2.1 ± 0.6	NA	4 ± 1.0	5.4 ± 1.5	5.4 ± 1.1	6.2 ± 3.8
Coated panel	RInb (%)	NA		NA		NA	
	TEQ	NA		NA		NA	
Uncoated panel	RInb (%)	NA		NA		NA	
	TEQ	NA		NA		NA	
Blank	RInb (%)	NA		NA		NA	
	TEQ	NA		NA		NA	

NA: no activity; RInb (%): relative inhibition in percentage; TEQ: Tamoxifen equivalent: μg Tamoxifen eq./dm^2^.

**Table 4 toxics-11-00156-t004:** Data obtained by LAB D and C for Anti-AR CALUX.

Assay	Data	Anti-AR					
Sample		LAB A		LAB B		LAB C	
Lab Test		D	C	D	C	D	C
Positive control	RInh (%)	77.2 ± 3.2	71.5 ± 6.7	71.6 ± 12.8	74.5 ± 7.9	69.7 ± 6.0	66.5 ± 12.9
	FEQ	16.7 ± 5.1	96.6 ± 103.6	31.7 ± 6.4	58.4 ± 14.7	40.3 ± 13.9	92.6 ± 26.1
Coated panel	RInh (%)	73.1 ± 7.7	NA	70.2 ± 11.0	NA	75.3 ± 1.7	NA
	FEQ	32.3 ± 3.8	NA	24.0 ± 4.2	NA	31.7 ± 8.5	NA
Uncoated panel	RInh (%)	NA		NA		NA	
	FEQ	NA		NA		NA	
Blank	RInb (%)	NA		NA		NA	
	FEQ	NA		NA		NA	

NA: no activity; RInb (%): relative inhibition in percentage; FEQ: Flutamide equivalent: μg Flutamide eq./dm^2^.

**Table 5 toxics-11-00156-t005:** Data obtained by Lab D and C for CALUX AhR (PAH).

Assay	Data	AhR					
Sample		LAB A		LAB B		LAB C	
Lab Test		D	C	D	C	D	C
Positive control	RI (%)	21.0 ± 3.0	19.8 ± 2.4	13.4 ± 3.0	13.9 ± 5.2	NA	NA
	B(a)PEQ	120 ± 42.4	60 ± 21.0	107 ± 41.6	91 ± 49	NA	NA
Coated panel	RI (%)	NA	12.4 ±3.9	14.1 ± 3.7	NA	NA	NA
	B(a)PEQ	NA	27.4 ± 15.7	120 ± 28.3	74 ± 89.7	NA	NA
Uncoated panel	RI (%)	33.2 ± 7.2	18.5 ± 4.4	NA	15.0 ± 15.9	NA	
	B(a)PEQ	320 ± 28.3	189 ± 156.8	NA	97 ± 42.3	NA	
Blank	RI (%)	13.4 ± 7.8	NA	NA	15.3 ± 2.8	NA	
	B(a)PEQ	80 ± 56.6	NA	NA	28 ± 32.9	NA	

NA: no activity; B(a)PEQ: B(a)P equivalents): ng B[a]P eq./dm^2^.

**Table 6 toxics-11-00156-t006:** Reference compounds (mean) performance as compared to standard ranges.

Guideline/Historical Data	Endpoint	Range (M)	LAB C (M)	LAB D (M)
TG458 ^1^	anti-AR	IC_50_: 1.1 × 10^−7^ to 1.1 ×10^−6^	9.2 × 10^−7^	4.8 × 10^−7^
TG455 ^1^	anti-ERa	IC_50_: 7.6 × 10^−9^ to 7.6 × 10^−8^	2.4 × 10^−8^	1.7 × 10^−8^
BDS ^2^	AhR	EC_50_: 1.6 × 10^−9^ to 1.6 × 10^−8^	3.3 × 10^−9^	7.2 × 10^−9^

^1^ OECD guidelines; ^2^ BioDetection Systems historical data.

**Table 7 toxics-11-00156-t007:** Summary BlueScreen™ HC Gadd45α-Gluc genotoxicity screening results in the absence and presence of S9 for prepared samples from LAB A, B and C analyzed by LAB B or C.

Assay	Gadd45α (−S9) (RI)						Gadd45α (+S9) (RI)					
Sample	LAB A		LAB B		LAB C		LAB A		LAB B		LAB C	
Lab Test	B	C	B	C	B	C	B	C	B	C	B	C
Positive control	NI		1.80 ± 0.31	1.27 ± 0.07	2.22 ± 0.15	1.24 ± 0.08	1.64 ± 0.34	1.30 ± 0.25	2.58 ± 0.16	1.58 ± 0.17	1.80 ± 0.33	1.60 ± 0.05
Coated panel	NI		NI		NI		NI		1.95 ± 0.31	1.07 ± 0.35	NI	
Uncoated panel	NI		NI		NI		NI		NI		NI	
Blank	NI		NI		NI		NI		NI		NI	

NI: no induction observed. RI: relative induction. Grey boxes indicate positive call according method criteria.

**Table 8 toxics-11-00156-t008:** Summary results from a 24-well microplate miniaturized Ames test for TA98 and TA100 strains in the absence (−) and presence (+) of S9.

Sample Type	TA98		TA100	
	(−) S9	(+) S9	(−) S9	(+) S9
Control Panel	-	-	-	-
Uncoated Panel	-	-	-	-
B(a)P	-	+	-	+
Process Blank	-	-	-	-
Ames test positive control	+ NF	+ 2-AA	+ SA	+ 2-AA

NF: nitrofluorene. 2-AA: 2-aminoanthracene. SA: sodium azide.

## Data Availability

Not applicable.

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
