# Peer review of "Interlaboratory Study to Evaluate a Testing Protocol for the Safety of Food Packaging Coatings"

_toxics, 2023, doi:10.3390/toxics11020156_

Round 1

Reviewer 1 Report

Comments on TOXICS 217866

The paper is excellent and merits the publication. There are only a few minor comments and/or recommendations that should be answered/added/corrected before final acceptation. The list follows:

1.- DCM injected in LC-HRMS? Please clarify, as DCM is not compatible with C18 stationary phase in LC.

2.- The classical liquid extraction using three sequential times 100 ml and later evaporate them to 1 mL is not environmentally friendly. As recommendation for the future, the method should be refined to decrease the organic solvent as much as possible.

Reviewer 2 Report

Interesting and rather thorough paper. Only two minor comments/requests:

L160: How were the laboratories selected? What was the criteria, e.g. were they accredited or..?

L190, L219, L295: Please explain what all the analysis are expected to find - e.g. SVHC, that would help reader in determining the outcome of the results better.

Reviewer 3 Report

The manuscript tested the intra and inter-laboratory reproducibility of a testing protocol for a can coating with regard to food safety.

The title seems to promise a safety assessment (“Interlaboratory Study to Assess the Safety of Packaging Materials…”), but finally, in line 731, it is clearly stated “The study aim was not to assess the safety of the material tested but to demonstrate the critical steps…”, more clearly than in the introduction. Also in line 21, “A protocol was developed for the chemical and biological screening of migrants from coated metal packaging materials”, which is not really true: a certain protocol was tested on reproducibility. It came not only as a disappointment to me as reader, but is also strange in as far as a protocol is tested of which it has not been demonstrated that it is fit for purpose. This probably sounds too harsh, since the protocol is an advanced one, perhaps one of the best. However, the question to what extent it demonstrates the safety of the migrate from the coating chosen remains open. There is no data on migration and the sensitivity of the tests for the detection highly toxic constituents in the mixture. Hence, the level of safety that the protocol can achieve remains open. For instance, the migrate probably included BADGE, which is Ames-positive, but was not detected by the described procedure. This should be clarified, as well as why the authors have chosen this protocol (which can probably be done by references).

The meaning of positive control is unclear

47: this sentence cannot be left as it is. Safe is safe and, hence, data requirements are basically the same for IAS as for NIAS, as clearly expressed by the EFSA opinion from 2016 (ref. 21). The only difference according to this opinion: For potential genotoxics, the TTC of 0.15 ppb is accepted.

53: the reference to Grob et al. is missing. An older paper (Food Additives and Contaminants 16 (1999) 579-590) shows for can coatings that the proportion of the NIAS may be more than 95 %.

62: Again not acceptable as it stands: it is not the legislator to specify what are the internationally recognised scientific principles on risk assessment. This is the up to EFSA, and EFSA has done this. How this should be achieved technically is up to industry. It is another point, whether this is technically feasible of not (and could be addressed in the discussion by the gap between the feasibility and the level of safety required by EFSA).

65: sentence is repeated

97: here it should be mentioned that the Ames test was developed for single substance testing and has a limited sensitivity for detecting minor constituents in a mixture.

157: PC seems to have two meanings. As I understand, a non-specified coating was used for the study (148) and the PC was only a “positive control” (whatever that meant).

178: justification to use 50 % ethanol as simulant?

203: Thermo in Switzerland?

453: Table 1 is for the CT samples?

468: the 9/26 is not in Table 1.

557: is there a reference to “LOQ criteria to consider the sample as antagonistic”? Same in line 576.

567-568: where are the thresholds from? Same in 680 and 688

Why is Table 3 needed?

680: is there a reference for the threshold 1.8?

In all, it is interesting work. I recommend publishing after some clarifications.

Round 2

Reviewer 3 Report

My points have been considered.